# SHED: Shapley-Based Automated Dataset Refinement for Instruction Fine-Tuning

**Yexiao He**[1]    **Ziyao Wang**[1]    **Zheyu Shen**[1]    **Guoheng Sun**[1]    **Yucong Dai**[2]
**Yongkai Wu**[2]    **Hongyi Wang**[3]    **Ang Li**[1]
[1]University of Maryland    [2]Clemson University    [3]Rutgers University
{yexiaohe,ziyaow,zyshen,ghsun,angliece}@umd.edu
{yucongd,yongkaw}@clemson.edu
hongyi.wang.001@rutgers.edu

## Abstract

The pre-trained Large Language Models (LLMs) can be adapted for many downstream tasks and tailored to align with human preferences through fine-tuning. Recent studies have discovered that LLMs can achieve desirable performance with only a small amount of high-quality data, suggesting that a large portion of the data in these extensive datasets is redundant or even harmful. Identifying high-quality data from vast datasets to curate small yet effective datasets has emerged as a critical challenge. In this paper, we introduce SHED, an automated dataset refinement framework based on Shapley value for instruction fine-tuning. SHED eliminates the need for human intervention or the use of commercial LLMs. Moreover, the datasets curated through SHED exhibit transferability, indicating they can be reused across different LLMs with consistently high performance. We conduct extensive experiments to evaluate the datasets curated by SHED. The results demonstrate SHED's superiority over state-of-the-art methods across various tasks and LLMs; notably, datasets comprising only 10% of the original data selected by SHED achieve performance comparable to or surpassing that of the full datasets.

## 1 Introduction

The development of LLMs marks a major leap in machine learning, transforming how we approach natural language processing (NLP) and artificial intelligence (AI) research [1, 2, 3, 4, 5]. LLMs such as GPT-3 [2], Mistral [6], and LLaMA/LLaMA2 [3, 4] highlight the benefits of pre-training on large and diverse mixtures of data corpora, empowering these LLMs with a wealth of knowledge[7, 8]. Moreover, one of the pivotal strengths of LLMs lies in their adaptability to specific tasks through fine-tuning. Fine-tuning, a process that involves adapting LLMs to one or multiple task-specific datasets, enables the pre-trained LLM to acquire task-specific information. Furthermore, it facilitates the alignment of LLMs to more accurately follow human instructions through fine-tuning on a dataset comprised of instructions paired with appropriate responses[9], which is known as instruction tuning.

However, fine-tuning LLMs also raises challenges. A primary concern is that noisy data or harmful instances in the fine-tuning dataset can significantly degrade the performance of pre-trained LLMs [10]. While many works have developed large and diverse datasets for fine-tuning purposes, recent research suggests that meticulously curated datasets of high quality, even if smaller in size, can be more effective in harnessing the full potential of LLMs [11, 12, 13]. Indiscriminately increasing the volume of data can lead to ineffective performance improvements and might even deteriorate LLM performance due to the introduction of noisy and harmful instances. Additionally, for instruction tuning, the LLM has already learned the necessary knowledge in the pre-training stage. The dataset used in the fine-tuning stage merely aims to better align the LLM to follow human instructions, indicating that this process does not necessitate extensive data [14]. Furthermore, fine-tuning LLMs on extensive datasets incurs significant computational costs. The necessity for considerable GPU

38th Conference on Neural Information Processing Systems (NeurIPS 2024).

resources presents a critical challenge [15]. Only researchers and institutions equipped with sufficient computing resources can perform such tasks, limiting broader applications and progress within the LLM community. Consequently, there is a pressing need to design a novel method for curating small and high-quality datasets that enable efficient fine-tuning.

Previous efforts have employed various methods such as curation or generation through manual efforts or commercial LLMs [11, 16], identifying subsets from larger datasets via training dynamics or estimating marginal contributions [17, 18]. Most current methods for data selection neglect the potential influence that different combinations of samples can have on model performance. The Shapley value [19], introduced in cooperative game theory, provides a method for fairly evaluating the contribution of each participant by examining all possible combinations and their effects on the overall result. This principle has also been utilized in machine learning to assess the impact of individual data points within a given dataset [20]. The Shapley value can serve as a criterion to refine one or more large datasets to extract high-quality data points, enabling the curation of a smaller yet high-quality dataset. This method not only facilitates the selection of impactful data but also considers the effectiveness of selected data combinations. The Shapley value seems to be a promising tool for data selection. However, calculating the Shapley value for all the data samples in a dataset is computationally expensive, especially for large-scale fine-tuning datasets.

Motivated by the aforementioned challenges, we present SHED, a Shapley-based automated dataset refinement framework for fine-tuning LLMs. The key intuition behind SHED is to perform Shapley value evaluations on a small portion of representative samples only, thereby dramatically decreasing the computational complexity of Shapley-based data refinement.

Specifically, as Figure 1 illustrates, SHED consists of three key components: *(1) model-agnostic clustering*, *(2) proxy-based Shapley calculator*, and *(3) optimization-aware sampling*. Initially, the model-agnostic clustering groups embeddings of the original dataset and then selects representative data samples as a proxy for each

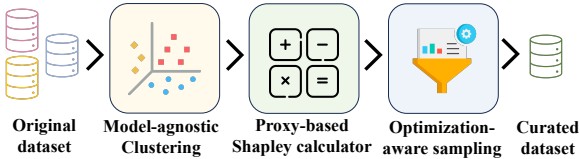

Figure 1: Overview of SHED.

cluster based on the distance of embeddings to the cluster centroid. These proxy data instances are then evaluated by the proxy-based Shapley calculator, which employs an approximation method to efficiently calculate their Shapley values, focusing on task-specific objectives (*e.g.,* accuracy and fairness). This method involves iteratively removing groups of instances from the proxy dataset and assessing the performance variation of the model to estimate the collective contribution of these instances, thereby streamlining the computation of Shapley values. The derived Shapley values of these proxy data instances are used as the quality score for their respective clusters. Finally, optimization-aware sampling selects data from clusters to compile a compact yet high-quality dataset, employing strategies that may favor clusters with higher-quality scores.

SHED only computes Shapley values for the cluster representatives rather than each data point, drastically boosting the efficiency of data refinement. Furthermore, Yang et al. (2022) observed that hyperparameters tuned on smaller models can be effectively transferred to larger models, significantly reducing tuning costs while maintaining performance [21]. We observed a similar phenomenon: datasets curated by SHED exhibit strong transferability, performing robustly across LLMs of various sizes and families. This suggests that smaller LLMs can be used for data selection, reducing computational costs. The selected datasets can be used to fine-tune larger LLMs and reused in multiple tasks to further amortize costs. Moreover, SHED offers a unified yet flexible framework, catering to various user needs by providing multiple options within each component. For example, the optimization objective for Shapley value measurement can be tailored to specific tasks (*e.g.,* fairness). Our key contributions can be summarized as follows:

- We present SHED, a generic data refinement framework based on Shapley values, which can curate a small yet high-quality dataset for boosting the efficiency of fine-tuning LLMs.
- We conducted extensive experiments on two benchmark datasets, *i.e.,* MMLU and WizzardLM, the results demonstrate that fine-tuning LLMs with small datasets curated by SHED yields performance comparable to, or even better than, using the original large datasets. Notably, datasets curated by SHED exhibit strong transferability, achieving robust performance across various LLMs of different sizes and families. This indicates that smaller models can employed to greatly lower computational expenses for data selection, and the

selected dataset can be used to fine-tune larger models and reused across multiple tasks to further distribute the costs.

- Code associated with the collection of high-quality datasets curated by SHED can be found at SHED: Shapley-Based Automated Dataset Refinement.

## 2 Related Work

### 2.1 Coreset Selection

Coreset selection plays a critical role in machine learning by targeting the selection of a representative subset from a larger dataset. Various coreset selection methods use unique criteria for choosing samples. Geometry-based approaches focus on the geometric properties of the data points, striving to retain geometrically significant samples that represent the overall data distribution [22, 23, 24, 25]. Uncertainty-based methods choose samples based on the uncertainty they present to the model, typically engaging samples that the model finds challenging to classify [26, 27, 28]. Decision-boundary-based methods select samples that are close to the decision boundary of the classifier, ensuring that the nuances of the classification boundary are well-represented in the selected subset [29, 30]. Gradient-matching approaches involve selecting a subset that yields similar gradient distributions as the entire dataset when used in training [31, 32]. Bilevel Optimization optimizes the coreset selection in a way that the selected subset maximizes certain performance metrics [33]. Dataset Selection with Datamodels using datamodels to approximate how the learning algorithm utilizes different subsets of training data to minimize target task loss.[34, 35] Submodularity-based approaches consider both diversity and information richness, striving for a balanced representation of the dataset [36].

### 2.2 Data Selection for Instruction Fine-tuning

Due to the superiority of instruction fine-tuning in enhancing the performance of LLMs, many recent studies focus on selecting high-quality instruction fine-tuning data. Based on methods, it can be divided into the following categories. Indicators-based methods define multiple metrics, such as instruction length and perplexity, to compute quality scores for each instruction instance [16, 37, 38, 39]. Training-based methods leverage the performance improvement through fine-tuning to score and select instruction data suited for fine-tuning [18, 40, 41, 42, 43, 44, 45]. Some other methods employ commercial LLMs like ChatGPT to assess quality, complexity, and diversity of instructions for selection [13, 46, 47, 48, 49].

### 2.3 Limitations of Previous Work

Most existing methods for data selection overlook the impact of various data combinations on model performance. As Table 1 illustrates, datasets formed by combining high-quality data, which are merely based on the independent quality score of each individual data sample, do not necessarily enhance model performance effectively. The combination of different data can impact the final performance of fine-tuning.

Although TS-DSHAPLEY [18] also utilized Shapley value for data selection, SHED offers several distinct advantages. SHED computes Shapley values only for proxy data of clusters rather than each individual data point, dramatically reducing computational overhead compared to TS-DSHAPLEY. SHED employs model-agnostic clustering, enhancing the transferability of curated datasets across different language models and model families. Moreover, SHED considers data diversity and can be customized for various optimization objectives, while TS-DSHAPLEY primarily focuses on predictive accuracy.

Many other existing works are also task-specific, limiting their applicability. In contrast, SHED offers a unified and flexible framework, adaptable to various instructional tuning tasks, making it more widely applicable.

## 3 Proposed Method

Motivated by the aforementioned challenges, we present SHED, a generic framework that exploits Shapley value to identify and select high-quality data to improve the performance and efficiency of fine-tuning LLMs.

### 3.1 Preliminary

The motivation behind this work is underscored by the observation, as illustrated in Table 1, that naively aggregating high-quality data merely based on the independent importance of individual

Table 1: We apply DSIR [50] to compile a high-quality dataset (10k instances), a random dataset (10k instances) from MMLU, and a mixed dataset samples 5k instances from each of the high-quality and random datasets. We fine-tune the LLaMA-7B model [3] on the curated dataset and evaluate them using the MMLU test set.

| Dataset | High-quality | Random | Mixed |
|---------|--------------|--------|-------|
| MMLU | 40.04 | 39.13 | 40.92 |

samples does not guarantee a performance improvement of fine-tuning. We believe this phenomenon is attributed to the complex interactions between different instances within the fine-tuning process. Thus, there is a pressing need to design a novel data selection method, which accounts for the individual and collective contributions of instances to model performance.

The Shapley value offers a compelling solution to this challenge. It quantifies the marginal contribution of each instance to the overall performance of the model, considering all possible combinations of instances. The formulation of the Shapley value for a data sample $i$ in dataset $D$ can be expressed as:

$$S_i = \sum_{P \in D \setminus \{i\}} \frac{|P|!(|D| - |P| - 1)!}{|D|!} (v(P \cup i) - v(P)), \tag{1}$$

where $S_i$ is the Shapley value of $i$, $P$ is the subset of dataset $D$, $|D|$ and $|P|$ are the total number of instances in $D$ and $P$, $v(P)$ is the value function of $P$, which represents the performance of the LLM model fine-tuned on the subset $P$. As Eq. 1 indicates, the Shapley value of an instance $i$ captures its average impact on model performance across all subsets it might be part of. This ensures a fair evaluation of the contribution of each instance in the original dataset, enabling the selected data is genuinely beneficial for enhancing model performance when integrated with other data samples.

Additionally, the value function $v(P)$ in Eq. 1 serves to calculate contributions from corresponding data. This value function can be tailored for various optimization objectives, such as accuracy and fairness, facilitating the selection of data that aligns with the task-specific requirements.

However, computing the Shapley value, as depicted in Eq. 1, demands extensive computational efforts, because it requires evaluating the contribution of each instance across all possible combinations. For a dataset with $|D|$ instances, there are a total of $2^{|D|} - 1$ possible combinations. For each combination, two evaluations are needed, *i.e.,* one includes a certain instance and the other one holds out that instance, doubling the computational workload to determine the contribution of that particular instance. Thus, the time complexity for measuring the Shapley value of each instance is $O(2^{|D|})$. Given the need to perform this calculation for all $|D|$ instances to determine their individual Shapley values, the overall time complexity for the dataset increases to $O(|D| \cdot 2^{|D|})$. This exponential complexity makes direct computation of Shapley values impractical for large datasets.

### 3.2 Design of SHED

To address the above challenges, we design SHED, comprising of three key components: model-agnostic clustering, proxy-based Shapley calculator, and optimization-aware sampling. We introduce each component in detail.

**Model-agnostic Clustering.** Given the time complexity of computing the Shapley value, calculating the Shapley value for all instances in a large fine-tuning dataset is impractical. The model-agnostic clustering employs models from Sentence Transformers [51] to generate semantically meaningful embeddings for each sample in the original dataset. These embeddings facilitate the efficient and effective computation of semantic similarities between textual inputs, enabling the grouping of data with similar contexts. Moreover, those model-agnostic embeddings enhance the transferability of the curated dataset, as demonstrated in Table 7. Then, the model-agnostic clustering applies algorithms, such as K-means [52] and Agglomerative Clustering [53], to group the embeddings. It then selects the representative data, which is closest to the cluster centroids in the embedding space, for each cluster. In doing so, we use these representative samples as the proxy of the respective clusters. Subsequently, SHED only calculates the Shapley values of those proxy data, using their Shapley values as the quality scores for their respective clusters. Employing proxy data effectively captures the essence of the diversity and complexity in the dataset. This strategy significantly reduces the computational burden associated with calculating Shapley values across vast datasets.

**Proxy-based Shapley Calculator.** To further improve efficiency for Shapley value calculations, the proxy-based Shapley calculator employs an approximation method to estimate the Shapley values of the proxy data. This method iteratively removes groups of $n$ instances from the proxy data $D_p$, followed by an evaluation of the model's performance to assess the impact of these instances. The performance variations before and after the removal of a specific group of instances quantify their collective contribution. Specifically, the contribution of the initial group of $n$ instances, denoted as $c_{(1..n) \in D_p}$, is computed by $c_{(1..n) \in D_p} = v(D_p) - v(D_p \setminus \{1..n\})$. Similarly, the contribution for the subsequent group of $n$ instances is determined by $c_{(n+1..2n) \in D_p} = v(D_p \setminus \{1..n\}) - v(D_p \setminus \{1..2n\})$. This procedure is repeated, progressively removing groups of $n$ instances until the entire proxy data has been visited, which marks the completion of a single iteration. This entire iteration process is then repeated $k$ times to enhance the accuracy of the approximation. After completing $k$ iterations, the Shapley value for a certain instance $i$ of the proxy dataset is approximated using the average of its contributions across all iterations, defined as $S_i \approx \frac{1}{k} \sum_k \frac{c_i(k)}{n}$, where $c_i(k)$ denotes the contribution associated with instance $i$ in the $k$th iteration.

**Optimization-aware Sampling.**
The Shapley value of each proxy data is assigned as the quality score of the corresponding cluster. Optimization-aware sampling utilizes these quality scores to sample data from these clusters, aiming to curate a small yet high-quality dataset. Optimization-aware Sampling offers two sampling methods: Quality-Ordered Cluster Sampling (QOCS) and Quality-Weighted Cluster Sampling (QWCS). QOCS prioritizes sampling from clusters with the highest quality scores. It selects instances starting from the most high-quality clusters until a pre-defined target sampling number is reached. QWCS adopts a prob-

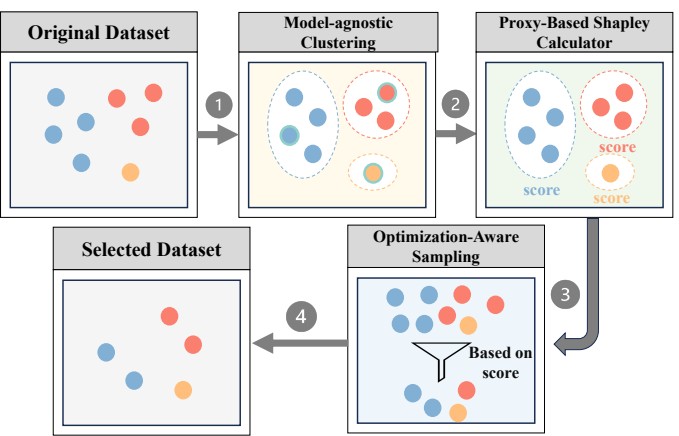

Figure 2: Workflow of SHED: ① Clustering and determining proxy data; ② Calculating Shapley values as scores; ③ Sampling based on scores; and ④ Forming the selected dataset.

abilistic approach to sample instances across all clusters, with the probability of selection from a given cluster weighted by its quality score. This method aims to balance quality with diversity by allowing for the inclusion of instances from a broader array of clusters, thus potentially enriching the dataset with a wider variety of high-quality data points. The probability $\Pr(i)$ of selecting an instance from cluster $i$ is defined in Eq. 2:

$$\Pr(i) = \frac{e^{fS_i}}{\sum_i e^{fS_i}}, \tag{2}$$

where $S_i$ represents the quality score of cluster $i$, and $f$ is a scaling factor that modulates the emphasis on quality versus diversity within the sampled dataset. By adjusting $f$, users can tailor the sampling process to prioritize either quality or diversity to suit specific task goals. A higher $f$ value tends towards selecting higher-quality instances, offering a versatile toolkit for dataset optimization.

## 4 Experiments

### 4.1 Experimental Setup

**Datasets.** We conduct experiments on two famous benchmark datasets, MMLU (99.8k instances) [54] and WizardLM-evol-instruct-70k (70k instances) [55].

**SHED Implementation.** We use the K-means algorithm for the model-agnostic clustering and set the number of clusters to 3000. For the proxy-based Shapley calculator, the value function is set as the accuracy of the foundation model fine-tuned on the proxy data. We use LLaMA-7B [3] as the

pre-trained foundation model and 10% instances in the MMLU test set calculating the Shapley values of proxy data. The number of iterations $k$ is set to 10, and the number of instances $n$ removed from the proxy data each step is set to 60. To conserve time and resources, instruction fine-tuning within the proxy-based Shapley calculator is conducted for one epoch. For optimization-aware sampling, we employ the QOCS and QWCS strategies with setting the scaling factor to 1, investigating their efficacy with a variety of target sampling sizes. These implementations are denoted as **SHED-QOCS** and **SHED-QWCS**. The target sampling size varies from $1,000$ to $20,000$ with increments of $1,000$, to thoroughly assess the impact of each sampling approach on fine-tuning performance.

**Baseline Methods.** We compare SHED with three baseline methods. Specifically, we implement a random-sampling method, denoted as **RS**, which randomly selects a subset from a large dataset. We also use the Dataset Quantization method [38], denoted by **DQ**, and the Data Selection with Importance Resampling [50], denoted by **DSIR**, for comparisons. In addition, we also consider fine-tuning models on the entire dataset, denoted as **FULL**, as a baseline.

**Evaluation Settings.** After obtaining the curated datasets using SHED and baseline methods, we fine-tune the pre-trained models using each curated subset, respectively. We apply the Low-Rank Adaptation (LoRA), which is a flexible and efficient tool, for fine-tuning and set the default LoRA rank to 128 [56, 57]. For all curated datasets, the instruction fine-tuning was conducted for 3 epochs. Notably, we use the same hyperparameters in fine-tuning across all methods to ensure a fair comparison, aiming to isolate the impact of data selection on model performance. We evaluate the performance of fine-tuned models on MMLU and ARC-challenge tasks using the `lm-evaluation-harness` testing framework [58]. To better evaluate the human preferences of fine-tuned models, we adopt `MT-Bench` [59] in our experiments. All the experiments are conducted on two A100 GPUs, each with 80GB of memory.

## 4.2 Experiment Results

We summarize the experimental results for SHED and other baseline methods. For consistency, the **bold** numbers indicate the corresponding method outperforms the **FULL** method. Additionally, we underline the best result achieved among all the methods that curate subsets.

For each method, the dataset from the curated collections that yields the optimal result across various sample sizes is referred to as the **best-selected dataset**.

Table 2: Performance comparison of curated datasets of the same size by SHED and baseline methods.

| Original dataset | MMLU | | | | | WizardLM | | | | |
|---|---|---|---|---|---|---|---|---|---|---|
| Method | RS | DQ | DSIR | QOCS | QWCS | RS | DQ | DSIR | QOCS | QWCS |
| MMLU | 38.94 | 39.88 | 40.24 | 44.80 | 43.87 | 33.12 | **33.20** | **33.86** | 35.43 | **34.91** |
| ARC-challenge | 45.10 | **46.35** | 45.67 | **47.10** | **47.23** | 46.01 | **48.71** | 47.66 | **49.47** | **49.92** |

Table 3: Performance of the best-selected datasets of SHED and baseline methods on the MMLU task.

| | MMLU | WizardLM |
|---|---|---|
| QOCS | 44.80 (10k) | **35.92** (4k) |
| QWCS | 44.24 (13k) | **35.76** (9k) |
| RS | 40.87 (15k) | **34.33** (7k) |
| DQ | 43.50 (7k) | **33.97** (7k) |
| DSIR | 40.23 (13k) | **34.72** (10k) |
| Full | 45.56 (99.8k) | 33.16 (70k) |

**Effectiveness of SHED.** Given the datasets generated from SHED and the baseline methods, we fine-tune the LLaMA-7B model, respectively, and evaluate the fine-tuned models on the MMLU and ARC-challenge tasks. We compare the results of the datasets of 10k instances curated by SHED and the baseline methods. As depicted in Table 2, when the number of total sampling instances is fixed (10k), the datasets curated by SHED consistently outperform those chosen by baseline methods. We also compare the performance of fine-tuned models using the best-selected dataset by each method. Table 3 shows the evaluation results for the MMLU task. Our method, **SHED-QOCS**, demonstrated superior performance on the MMLU dataset compared to baseline methods, achieving the highest results among the curated datasets. Furthermore, **SHED-QOCS** also led in performance

Table 4: Performance of the best-selected datasets of SHED and baselines on the ARC-challenge task.

|  | MMLU | WizardLM |
|---|---|---|
| QOCS | **47.10** (10k) | **51.36** (1k) |
| QWCS | **49.21** (9k) | **50.26** (7k) |
| RS | **47.07** (13k) | **49.33** (16k) |
| DQ | **46.50** (3k) | **50.24** (5k) |
| DSIR | **46.90** (3k) | **48.78** (12k) |
| Full | 45.99 (99.8k) | 47.95 (70k) |

Table 5: MT-Bench evaluation of the best-selected datasets of SHED and baselines.

| Original dataset | MMLU | | | | | WizardLM | | | | |
|---|---|---|---|---|---|---|---|---|---|---|
| Method | Full | RS | QOCS | RS | QWCS | Full | RS | QOCS | RS | QWCS |
| Size | 99.8k | 10k | 10k | 13k | 13k | 70k | 4k | 4k | 9k | 9k |
| LLaMA-7B | 3.02 | 2.23 | 2.53 | 2.44 | 2.83 | 5.21 | 4.77 | 4.89 | 4.81 | **5.24** |

when utilizing the WizardLM dataset. It is notable that **SHED-QOCS** outperforms the full dataset, achieving a 2.76% higher accuracy. In Table 4, we report the results of the ARC-challenge task. Similarly, among the datasets curated from the MMLU dataset, the selected dataset of our method **SHED-QWCS** achieves the best result compared with the baseline methods. It also surpasses the full dataset by 3.22%. Within the datasets derived from WizardLM, **SHED-QOCS** once again curated the dataset of best performance, which surpasses the full dataset by 3.41%. The results demonstrate the effectiveness of SHED. Although SHED demands more computational effort, its strength lies in creating high-performance datasets.

**Evaluations on MT-Bench.** We use MT-Bench to evaluate the performance of datasets curated by SHED in terms of human preferences. Table 5 demonstrates that the datasets curated by SHED align well with human preferences, not only enhancing accuracy but also enabling the model to better understand and follow human instructions, generating answers that are more favorable to humans. The dataset constructed through the SHED-QWCS method, sampled from WizardLM, achieved a remarkable score of 5.24 on the MT-Bench. The results presented in Table 5 represent the average of five independent runs.

**Transferability Evaluation of Curated Datasets across Various Models.** To evaluate the transferability of datasets curated by SHED, we first apply SHED to select data from the MMLU and WizardLM datasets based on LLaMA-7B. Then, we fine-tune LLaMA-13B, Vicuna-7B, and GPT-2 using the best-selected dataset curated by SHED and the baseline methods. As summarized in Table 6 and Table 7, datasets curated by SHED exhibit robust performance across various models, demonstrating their transferability and applicability across various tasks and even different model families. The strong transferability of the curated datasets indicates that SHED identifies generally high-quality data. The computational cost for data selection can be significantly amortized across various models. In addition, the datasets selected by LLaMA-7B also achieve good performance when fine-tuning the larger model LLaMA-13B. This indicates that we can utilize smaller models to select data, thereby significantly reducing the computational cost of data selection.

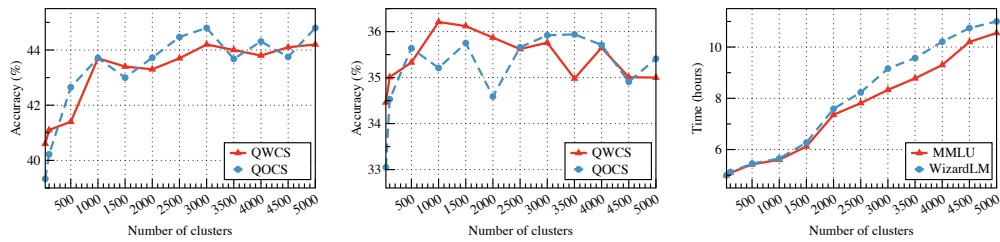

(a) Subsets selected from MMLU. (b) Subsets selected from WizardLM. (c) Computational time for one iteration of Shapley value calculation.

Figure 3: Performance of subsets with varying numbers of clusters in SHED.

Table 6: Transferability evaluation using the best-selected datasets across different models on MMLU task.

| Original dataset | MMLU | | | | | WizardLM | | | | |
|---|---|---|---|---|---|---|---|---|---|---|
| Method | Full | RS | QOCS | RS | QWCS | Full | RS | QOCS | RS | QWCS |
| Size | 99.8k | 10k | 10k | 13k | 13k | 70k | 4k | 4k | 9k | 9k |
| LLaMA-13B | 53.22 | 50.04 | 52.95 | 50.12 | 51.54 | 45.63 | 45.77 | **45.93** | 45.81 | **46.36** |
| VICUNA-7B | 49.70 | 48.43 | **50.01** | 47.21 | 48.93 | 45.56 | 45.71 | **47.19** | 45.44 | **48.16** |
| GPT-2 | 24.22 | 23.74 | **26.89** | **24.33** | **25.83** | 26.19 | 25.07 | **26.76** | 24.85 | 25.77 |

Table 7: Transferability evaluation using the best-selected datasets across different models on ARC-challenge task.

| Original dataset | MMLU | | | | | WizardLM | | | | |
|---|---|---|---|---|---|---|---|---|---|---|
| Method | Full | RS | QOCS | RS | QWCS | Full | RS | QOCS | RS | QWCS |
| Size | 99.8k | 10k | 10k | 13k | 13k | 70k | 4k | 4k | 9k | 9k |
| LLaMA-13B | 49.31 | 47.31 | **50.43** | 48.83 | **50.68** | 54.09 | 53.17 | **55.20** | 54.11 | **55.63** |
| VICUNA-7B | 44.88 | 44.86 | **45.23** | 43.24 | **44.91** | 49.91 | 47.72 | **50.26** | 47.98 | 48.72 |
| GPT-2 | 19.45 | 18.77 | **19.81** | 19.02 | **20.05** | 19.19 | 17.98 | **19.28** | 18.72 | **19.54** |

**Impact of Number of Clusters.** The number of clusters in K-means affects the computational cost needed for Shapley value calculations and the relevance of proxy data to its cluster. An increase in the number of clusters leads to smaller and more homogeneous groups, thereby improving the proxy data's representativeness for its respective clusters. However, this comes at the cost of increased computational overhead, highlighting a balance that must be struck to optimize both efficiency and representativeness. In this experiment, we evaluate the best-selected dataset by SHED across varying numbers of clusters using LLaMA-7B on the MMLU test set. Guided by the findings in [60], our investigation begins with a baseline cluster count of $C = \sqrt{|D|}$. We present the computation time for Shapley value computations across different settings, maintaining consistency with the experimental setup outlined in Section 4.1.

As Figure 3(a) and Figure 3(b) show, the results reveal that performance improvements of curated dataset reach a plateau when the number of clusters exceeds $3\sqrt{|D|}$. Meanwhile, Figure 3(c) demonstrates a proportional increase in computation time for Shapley value calculations as the number of clusters rises. Notably, at very low cluster counts (e.g., below 1000), Shapley value computation times are largely dictated by the evaluation, with the time spent remaining relatively constant across varying datasets. In such cases, the computation time is more significantly affected by the size of pre-trained models rather than the number of clusters itself. Given the transferability of datasets curated using the SHED, it is feasible to employ a smaller foundational model than the target model within the proxy-based Shapley value calculator. In doing so, the computation overhead for evaluation can be significantly reduced, making SHED a practical approach in real-world settings.

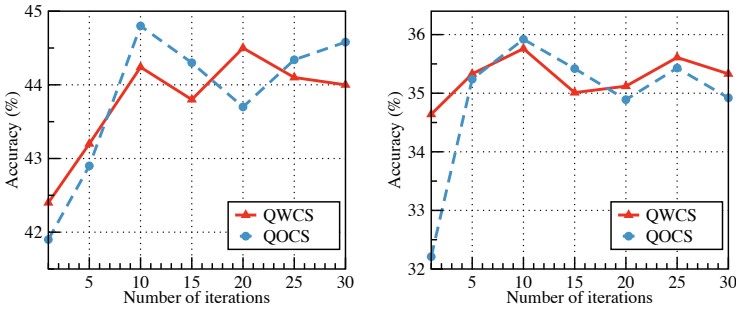

(a) Subsets selected from MMLU.  (b) Subsets selected from WizardLM.

Figure 4: Performance of subsets with varying iterations in SHED.

**Impact of Number of Iterations on Proxy-based Shapley Calculator.** The precision of Shapley value estimates increases with the number of iterations $k$, providing a more accurate measurement of each data sample's contribution to the model performance. However, this increment also leads to a proportional rise in computational cost, leading to a contrasting relationship between computational efficiency and the accuracy of Shapley value estimations. To seek the optimal number of iterations

for Shapley value calculations, we analyzed the performance of datasets curated by SHED under varying iteration settings. The experiments are conducted with the LLaMA-7B model on the MMLU test set, following the experimental settings detailed in Section 4.1.

Figures 4(a) and 4(b) illustrate that the performance of the curated datasets by **QOCS** and **QWCS** are stable once the iteration number surpasses 10. This result highlights the stability of our methods beyond 10 iterations, showing that further iterations beyond this threshold do not significantly improve dataset quality. Given the balance between computational cost and performance, setting the number of iterations to 10 is recommended for optimal efficiency and robustness.

## 5 Discussion

### 5.1 Data Selection for Multiple Tasks.

In our experiments, we thoroughly evaluate methods regarding accuracy. It is notable that our framework is readily adaptable. By setting different value functions $v(P)$, SHED can select any subset using arbitrary criteria. This adaptability allows SHED to customize its data selection process to produce a small dataset while improving specific objectives, such as model fairness [61].

In particular, if we aim to curate a dataset using the common fairness notion, *i.e.,* demographic parity, we can define $v(P)$ the disparity in positive prediction rates between groups with protected attributes (e.g., males vs. females), calculated as the negative absolute difference $-|X_{\text{Male}} - X_{\text{Female}}|$, where $X_{\text{Male}}$ and $X_{\text{Female}}$ are the positive prediction rates for male and female groups, respectively.

### 5.2 Complexity Analysis

We assume that the running time required to fine-tune the model using a single instance is denoted by $t$, and the time needed to evaluate the model on a test set consisting of $m$ instances is represented by $T_m$. Let $C$ denote the number of clusters, $n$ denote the number of instances within a group and $k$ signifies the number of iterations utilized in the proxy-based Shapley calculator as illustrated in Section 3.2. The total number of evaluations and fine-tuning per iteration would be proportional to $\frac{C}{n}$. For simplicity, we assume that $C$ is evenly divisible by $n$ for simplicity. Given $k$ iterations, the overall time complexity of this approximation method can be expressed as $\mathcal{O}\left(\frac{Ck}{n}\left[\frac{(C+n)t}{2} + T_m\right]\right)$.

## 6 Conclusion

In this work, we introduced SHED, an innovative Shapley value-based framework designed to refine datasets for the efficient fine-tuning of LLMs, addressing the computational hurdles commonly associated with Shapley value calculations through a novel clustering and proxy-based approach. Through extensive experiments conducted on benchmark datasets such as MMLU and WizardLLM, we have shown that LLMs fine-tuned with datasets curated by SHED not only match but, in some cases, surpass the performance of those trained with the original, larger datasets. Significantly, SHED-curated datasets have demonstrated a high degree of transferability, maintaining robust performance across various models. Furthermore, SHED's flexibility and efficiency underscore its potential to revolutionize LLM fine-tuning by allowing for the creation of compact, high-quality datasets.

## 7 Limitations

This research, while presenting significant advancements, encounters certain limitations that merit attention for future work. Firstly, the method's reliance on sufficiently representative embeddings may limit its applicability in real-world scenarios where such embeddings are unavailable or inadequate. Future work will explore ways to reduce this dependency for broader applicability. Secondly, the use of clustering and proxy data may overlook rare but important samples. Future research will focus on improving clustering methods to better capture these samples. Additionally, the framework's current objective focuses predominantly on model performance, which may inadvertently lead to model bias. This singular focus overlooks the equally important aspect of model fairness, crucial for ensuring that models perform equitably across diverse groups. Recognizing this, our framework is designed to be extensible and objective-agnostic, laying the groundwork for incorporating additional criteria. In subsequent research, we plan to integrate considerations of model fairness alongside performance.

## 8 Ethics Statement

In this work, we present SHED, a generic data refinement framework utilizing Shapley values, aimed at assembling a compact yet effective dataset to boost the efficiency of the fine-tuning process of

LLMs. This study carefully avoids ethical issues beyond standard AI concerns, leveraging properly cited publicly available Internet text data. This approach ensures adherence to ethical data use standards, reflecting our commitment to responsible research practices in the AI field.

## Acknowledgements

We thank the anonymous reviewers for their valuable insights and recommendations, which have greatly improved our work. This research has been graciously funded by NSF 2431611.

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

# Appendix

## A  Hyperparameter Settings and Experimental Configuration

In our experiments, we employed the following hyperparameters: the number of training epochs was set to 3, the batch size was 128, the LoRA rank (`lora_r`) was 128, and the LoRA alpha (`lora_alpha`) was 256. For clustering, when the number of clusters (C) was 3000, the number of samples removed per group ($n$) was 60; when testing the impact of different C values on performance, $n = \frac{C}{50}$. The number of iterations for Shapley value calculation ($k$) was 10, and the learning rate was $3 \times 10^{-4}$. Data preprocessing involved using the `sentence-transformers/all-MiniLM-L6-v2` model to generate semantically meaningful embeddings for each sample in the original dataset, followed by applying the k-means algorithm to cluster these embeddings.

## B  Comparison of the SHED-QOCS and SHED-QWCS

We compared the performance of datasets of varying sample sizes using SHED-QOCS and SHED-QWCS methods. For fine-tuning, we utilized the LLaMA-7B model. Other experimental configurations were aligned with the parameters detailed in Section 4.1.

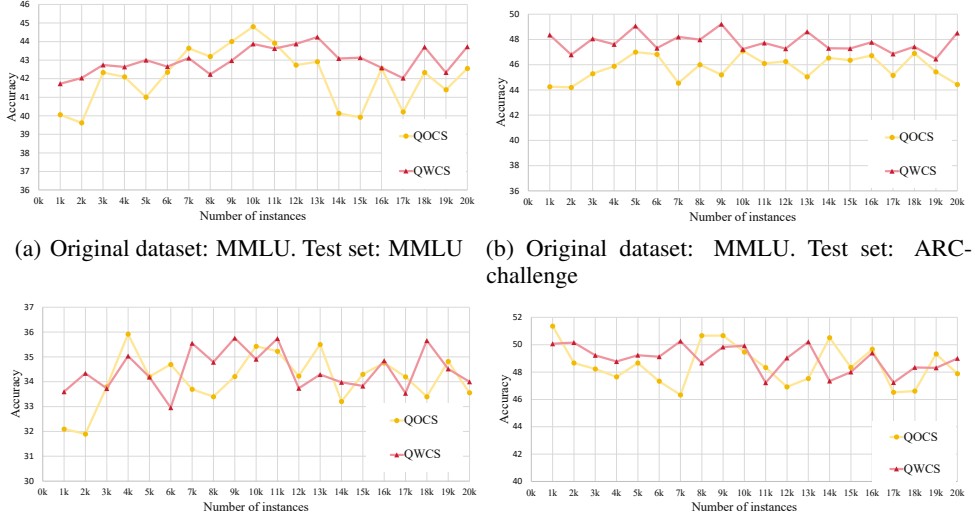

(a) Original dataset: MMLU. Test set: MMLU

(b) Original dataset: MMLU. Test set: ARC-challenge

(c) Original dataset: WizardLM. Test set: MMLU

(d) Original dataset: WizardLM. Test set: ARC-challenge

Figure 5: Results of curated datasets of different samples.

As figure 5 shows, datasets sampled using the SHED-QWCS approach generally outperform those obtained through SHED-QOCS, particularly with smaller sample sizes. This discrepancy is likely attributable to the limitation of SHED-QOCS in scenarios where the sample size is minimal. In such cases, SHED-QOCS tends to sample data from a limited number of clusters, leading to significant redundancy in the curated dataset.

Conversely, it is observed that datasets that achieve the best performance are often those sampled via SHED-QOCS. This improved performance is observed when the sample size is sufficiently large, allowing SHED-QOCS to sample data from a wider range of clusters. The inherent strength of SHED-QOCS lies in its strategic focus on harvesting high-quality data. As the sample size increases to a point where SHED-QOCS can effectively draw from multiple clusters, its advantage in prioritizing data quality becomes significantly beneficial.

Therefore, we suggest that users select between these two sampling methods based on the sample size they desire.

## C  Hyperparameter Tuning for Shapley Value Computation

To enhance user convenience, SHED introduces a method for setting hyperparameters.

Based on Figure 3(c), the time per iteration when calculating Shapley values exhibits an approximate linear relationship with the number of clusters. Similarly, the total time for computing Shapley values closely aligns linearly with the number of iterations. Thus, the computation time $t$ for Shapley values can be modeled as $t = \theta k C$. SHED randomly samples 2000 instances from the dataset to calculate the Shapley value for one iteration, recording this to determine $\theta$. To optimize $k$ to be close to 10 and the number of clusters near $3\sqrt{|D|}$, an optimization problem is formulated in Eq. 3.

$$\min_{k,C} \quad \lambda_1(k-10)^2 + \lambda_2(C - 3\sqrt{|D|})^2$$
$$\text{s.t.} \quad \theta k C = t_0$$

(3)

where $\lambda_1$ and $\lambda_2$ serve as weights, both defaulting to 1, while $t_0$ represents the maximum runtime set by the user. This optimization problem can be solved using the SQP (Sequential quadratic programming) method [62] to determine the optimal number of clusters and iterations.

## D  Comparison with Other Methods

We compare SHED with two methods, LIMA and IFD (using WizardLM) [12, 40]. LLaMA-7B is used as the base model. The results are presented below, focusing on performance in the MMLU and ARC-challenge tasks.

The following table shows the performance of SHED's selected dataset (with 1k samples) compared to LIMA's selected dataset (with 1k samples). It is important to note that SHED is fully automated, whereas LIMA involves manual curation.

| Task | SHED (1k) | LIMA (1k) |
|---|---|---|
| MMLU | 41.71 | 34.9 |
| ARC-challenge | 48.12 | 46.16 |

Table 8: Performance comparison between SHED and LIMA.

We compare SHED with IFD (with 7k samples) to select data from WizardLM. The following table shows the results.

| Task | SHED (1k) | SHED (7k) | IFD (7K) |
|---|---|---|---|
| MMLU | 33.62 | 35.63 | 33.08 |
| ARC-challenge | 51.36 | 50.11 | 52.90 |

Table 9: Performance comparison between SHED and Cherry-LLM (using WizardLM).

As the results demonstrate, SHED achieves competitive performance across both tasks, even with fewer samples.

