# OpenReview forum: "SHED: Shapley-Based Automated Dataset Refinement for Instruction Fine-Tuning"
_NeurIPS.cc/2024/Conference — NeurIPS 2024 poster_

### Official Review · Reviewer_tj1t · 2024-07-16

**Soundness:** 2
**Presentation:** 2
**Contribution:** 1
**Rating:** 3
**Confidence:** 4

**Summary:**

This paper explores the problem of Shapley-based data selection for instruction tuning. Specifically, the proposed approach is composed of three steps–clustering the target samples, Shapley-style evaluation for each clustering, and resampling from the clusters based on the Shapley scores. The paper validates its approach on two instruction tuning benchmarks and obtains favorable results against several baselines.

**Strengths:**

The paper is clearly written and very easy to read. The problem is well-contextualized and well-formulated.

The structure is complete and the language is neat.

Evaluations are standard and a number of baselines are considered.

**Weaknesses:**

The paper is probably a bit simple for a full publication of NeurIPS. It has concrete merits, though, I have concerns for several aspects of its contributions.

1. The paper adopts the expensive Shapley approach for data selection, citing its advantage for factoring in the combinatorial effects between samples. Yet, after estimating the Shapley score of the clusters, the final data selection is only conducted via resampling based on the score or ranking of the clusters, where the combinatorial effect between clusters is also not considered.

In this sense, I am not sure whether using the costly Shapley approach delivers actual benefits. A grid search over the sampling ratio from each cluster could have better results.

2. Instruction mining has been "extensively studied" during the past year and a wealth of papers have emerged after the paper [LIMA: Less is more for alignment]. Most of those papers claim to reach comparable performance using just a few percentages of the original instruction tuning dataset.

Yet, none of these papers are included as a baseline in this work. For the baselines compared within this work, DSIR is for pre-training data selection where its scalability and computation overhead and not at all comparable with the proposed method (and similarly for DQ).

3. It is very well known Shapley-style evaluations are notably expensive as they rely on repetitive model retrainings. It is important to benchmark the computation overhead for the proposed methods.

4. The paper claims the proposed method is suitable for any objectives. This very strong argument is not validated in this paper. For many tasks where this may not be true. For example, its computational complexity is prohibitive for pre-training data selection.
Also, this approach only applies to cluster-level data selection but not to sample-level selection. Its effectiveness seems highly dependent on the embedding space for clustering. This work uses Sentence-Transfomer in the experiments. It may not work for other tasks. At least, these major limitations are not discussed in this paper.

5. As mentioned before, a notable number of methods have been proposed for the problem of instruction mining. This paper does not clearly identify a research gap to fill. A similar Shapley-style approach has been proposed TS-DSHAPLEY. Experiments conducted in this paper are also limited. In general, I have concerns about the novelty and contribution of this paper.

**Questions:**

See Weaknesses.

**Limitations:**

The paper claims the proposed method is suitable for any objectives. This very strong argument is not validated in this paper. For many tasks where this may not be true. For example, its computational complexity is prohibitive for pre-training data selection.
Also, this approach only applies to cluster-level data selection but not to sample-level selection. Its effectiveness seems highly dependent on the embedding space for clustering. This work uses Sentence-Transfomer in the experiments. It may not work for other tasks. At least, these major limitations are not discussed in this paper.

---

> ### Author Rebuttal · Authors · 2024-08-07
>
> We thank the reviewer for the valuable feedback.
>
> **W.1:**
>
> We appreciate the reviewer's concern but note a possible misunderstanding about SHED.
>
> SHED does consider the combinatorial effects between clusters:
>
> - **Combinatorial Effects:** SHED calculates Shapley Value (SV) for cluster proxies, reducing computational cost while capturing inter-cluster relationships. Each cluster is represented by a proxy data point (closest to the centroid). **SVs are calculated by considering possible combinations of these proxies, thus considering interactions between clusters.** SV compresses the combinatorial space into a single score, encapsulating individual importance, synergies, and potential redundancies.
>
> - **Score-based Selection:** The selection step uses these scores, which already encapsulate combinatorial effects. **A high SV indicates a high likelihood of positive impact when combined with instances from other clusters.**
>
> Grid search for data selection incurs high costs.
> If there are C clusters and sampling ratio takes x values, with training time T. Grid search’s complexity is O(Tx^C), scaling exponentially with C, which is impractical.
>
> In contrast, Shed’s complexity is O((Ck/n)[(C+n)t/2 + T_m]). As Figure 3 shows, SHED can scale linearly with the number of clusters by setting C/n to a fixed value, making it feasible for large datasets. Thus, SHED offers a significant computational advantage over grid search.
>
>
> **W.2:**
>
> We appreciate the reviewer’s concern. To address this, we compared SHED with LIMA and Cherry-LLM:
>
> **LIMA:**
> - MMLU: SHED (1k) - 41.71 vs. LIMA (1k) - 34.9
> - ARC: SHED (1k) - 48.12 vs. LIMA (1k) - 46.16
> - Note: SHED is fully automated, unlike LIMA’s manual curation.
>
> **Cherry-LLM (using WizardLM):**
> - MMLU: SHED (1k) - 33.62, SHED (7k) - 35.63 vs. Cherry (7k) - 33.08
> - ARC: SHED (1k) - 51.36, SHED (7k) - 50.11 vs. Cherry (7k) - 52.90
>
> **SHED achieves comparable or better performance with less data.** Since SHED used MMLU as test set during data selection, its performance on ARC could be improved if used as test set during selection, showing SHED's effectiveness for task-specific data selection.
>
> Baseline Criteria:
>
> We use DSIR because it, like SHED, selects data by importance. DSIR samples data based on estimated importance, similar to SHED's use of SVs. We included DQ as it was also used for fine-tuning in its paper.
>
> We avoided methods requiring human or LLM evaluations due to high costs.
>
>
> **W.3:**
>
> We appreciate the reviewer’s concern about SHED's computational cost and would like to clarify:
>
> **Transferability & Cost Amortization:**
>
> - **One-time Computation for Multiple Uses:** SHED-refined datasets perform well across various model sizes and architectures. This transferability makes them reusable for different models or tasks. **This one-time computational investment ensures long-term efficiency, reducing the need for repeated data selection. SHED's amortized cost over several models significantly lowers the per-use cost.**
>
> - **Comparison with Non-transferable Methods:** Unlike methods requiring computation for each new model or task, SHED's transferability **eliminates recurrent computational expenses.**
>
> - **Efficiency Gains:** SHED-refined datasets lead to comparable or superior model performance with smaller datasets, directly translating to lower computational and time costs in the fine-tuning phase.
>
> **Efficiency Design:** SHED reduces the computational cost by approximating SVs for cluster proxies instead of individual data points. Our analysis and experiments show its complexity can grow linearly with the number of clusters, not exponentially with dataset size.
>
> **Benchmark:** Figure 3 (c) has shown the time for SV calculations across different numbers of clusters, which also indicates SHED’s scalability.
>
>
> **W.4:**
>
> We appreciate the reviewer’s comment but believe there may be a misunderstanding of our claims:
>
> - **Clarification of Objectives:** As highlighted in our paper, **SHED focuses on fine-tuning.** However, Figure 3 (c) shows linear growth in time, suggesting potential for large-scale pre-training data selection. SHED is adaptable to various **fine-tuning objectives**, such as task-specific performance, model fairness, and domain focus, by modifying the value function in SV. **This adaptability is intended within the fine-tuning context.**
>
> - **Not Pure Cluster-Level Selection:** While SHED uses clustering to reduce computational complexity, it does not solely rely on cluster-level selection. Instead, it considers individual samples within clusters. The final selection step, QWCS, allows for the selection of individual samples.
>
> - **Embeddings:** We use Sentence-Transformer, which is well-suited for NLP tasks and produces semantically meaningful embeddings. Trained on a large and diverse corpus, these embeddings are robust across various NLP tasks. SHED's framework can adapt to different tasks with suitable embeddings.
>
> **W.5:**
>
> We emphasized SHED's advantages over TS-DSHAPLEY in Section 2.3, including lower computational overhead, transferability, flexible framework, and data diversity.
>
> The unavailability of TS-DSHAPLEY code and datasets, along with significant computational demands, pose challenges for comparison. Nonetheless, the results presented in our paper demonstrate the superiority of SHED:
>
> - **Efficiency:**  Figure 3 (c) shows SHED's computational cost scales linearly with the number of clusters. **While TS-DSHAPLEY's cost scales with data size.**
>
> - **Transferability:** Tables 6, and 7 show SHED’s robust performance across different models, underscoring its transferability. TS-DSHAPLEY lacks evidence of effectiveness across varied models.
>
> - **Human alignment:** Table 5 shows SHED-refined datasets not only enhance accuracy but also align well with human preferences. TS-DSHAPLEY does not evaluate human preference alignment, leaving it unclear whether its selected datasets improve the model’s ability to follow human instructions.

---

> > ### Comment · Reviewer_tj1t · 2024-08-12
> > **Thanks for the responses.**
> >
> > Thanks for the responses. I have carefully read the rebuttal and other reviews. I appreciate the authors' effort, yet most of my opinions remain.
> >
> > I don't think there is a particular, critical flaw in the paper. It is a nice work that I would like to see in some venues, (e.g., as a short paper or at an ACL conference).  Given the existence of similar works such as TS-DSHAPLEY, I don't think it is in the best interest to publish it at NeurIPS.

---

> ### Author Response · Authors · 2024-08-12
>
> Thank you for carefully reading our rebuttal and providing your feedback. **We greatly appreciate your recognition that our paper does not contain any technical flaws. However, we are concerned about the score of 3, which is typically reserved for papers with significant technical issues or weak evaluations.** Since you acknowledged there are no critical flaws, a score of 3 seems inconsistent with the paper's merit.
>
> We would like to further clarify the distinctions and advantages of SHED compared to TS-DSHAPLEY:
> TS-DSHAPLEY randomly samples multiple subsets of the training data, estimates the contributions of data points in these subsets, and then repeats the process multiple times and aggregates the results to estimate the Shapley values for the entire training set. Notably, this method will calculate the Shapley value for all instances in the training set.
>
> Advantages of SHED:
> - **Improved computational efficiency:** SHED only computes Shapley values for representative proxy data samples selected from clusters, rather than for each individual data point as in TS-DSHAPLEY. This dramatically reduces the computational overhead, making SHED more scalable to large datasets.
>
> - **Model-agnostic data representation:** SHED uses model-agnostic sentence embeddings (e.g., from Sentence Transformers) to represent data samples, while TS-DSHAPLEY relies on representations extracted from the target language model. The model-agnostic approach enhances the transferability of the curated datasets across different language models, amortizing the computational cost of data selection. Table 7 showcases the transfer performance of SHED-curated datasets across different models. SHED is trained on the MMLU and WizardLM datasets using the LLaMA-7B model, and the resulting optimal subsets are used to fine-tune various models, including LLaMA-13B, Vicuna-7B, and GPT-2. The results demonstrate that SHED-constructed datasets exhibit robust performance across a range of models, confirming their applicability across models and even different model families. This strongly supports the transferability of SHED datasets.
>
> - **Clustering-based data sampling:** SHED employs clustering algorithms to group similar data samples and selects representative proxy data for each cluster. This helps capture the diversity and complexity of the original dataset while reducing redundancy. TS-DSHAPLEY does not explicitly consider data diversity in its sampling process.
>
> - **Flexible optimization objectives:** SHED's value function in the Shapley value calculation can be customized for various optimization objectives, such as accuracy and fairness, allowing the curated datasets to align with task-specific requirements. TS-DSHAPLEY focuses primarily on model predictive accuracy.
>
> - **Unified and extensible framework:** SHED presents a unified framework that integrates clustering, Shapley value computation, and data sampling, with the flexibility to accommodate different clustering algorithms, optimization objectives, and sampling strategies. This makes SHED more adaptable to various scenarios than TS-DSHAPLEY.
>
> We hope this clarification helps to further understand the unique contributions and strengths of SHED, and we kindly request reconsideration of the score in light of this information. **The existence of a related but less effective method should not be grounds for rejection.**
>
> Thank you once again for your feedback and consideration.

---

> > ### Comment · Reviewer_tj1t · 2024-08-13
> > **Thanks for the response.**
> >
> > Thanks to the authors for the response. Let me explain where I feel this work can be improved.
> >
> > "Model-agnostic data representation" relies on the quality of embedding space. Compared with random sampling, the effectiveness of clustering also depends on the embedding quality. Sentence-Transformer often works quite nicely despite being task-agnostic. This has been witnessed in many works. But on a larger picture, how we move forward from here remains unclear. On tasks where Sentence-Transformer cannot capture the nuances, what are we going to do?
> >
> > "Clustering-based data sampling" is a standard practice for reducing computation complexity when the scale of the problem becomes an issue. For example, [Gio: Gradient information optimization for training dataset selection] also implemented the idea of clustering-based data selection.
> >
> > The work strives for "Flexible optimization objectives" and "Unified and extensible framework". But the development of this paper concentrates on "selecting instruction samples". First, selecting instruction samples is not a particularly computationally intensive task such that many methodologies can be applied. There is now a wealth of paper on this topic. Since the overall size of these instruction-tuning datasets is typically not very large, there has been a constant debate on whether research should focus on "selecting a subset of instruction-tuning examples". Given this being a crowded field, the work could greatly improve its impact if extended to broader applications beyond "instruction mining".

---

> > > ### Author Response · Authors · 2024-08-13
> > > **Thank you for your feedback**
> > >
> > > Thank you for your thoughtful feedback. We appreciate the points raised, and would like to address each one in turn:
> > >
> > > 1. Embedding Quality and Model-Agnostic Data Representation:
> > > - While there may be edge cases where Sentence-Transformer does not fully capture nuances, **SHED’s flexible framework allows for integrating alternative or task-specific embeddings, which can be chosen based on the task’s particular needs.** For example, CodeBert can be used to replace the Sentence-Transformer for tasks involving code understanding. In addition, if a task requires nuanced domain-specific understanding, fine-tuning the Sentence-Transformer to adapt to these needs can be implemented, like LLaVA's method, which only requires a small cost. This adaptability ensures that SHED remains effective across a diverse set of tasks, even beyond the current scope.
> > > - **However, if one insists on a truly "perfect" task-agnostic model, even widely-used powerful models like LLaMA3 cannot claim to be suitable for every possible task. The task-agnostic nature we discuss is within a reasonable and practical range.**
> > >
> > > 2. Clustering-Based Data Sampling:
> > > 	While clustering is commonly used to reduce computational complexity, **SHED’s innovation lies in considering combinatorial effects between data points, model-agnostic data selection, reducing computation cost, and a flexible framework.**  Combining clustering with Shapley value calculations allows us to account for the combinatorial effects between data points, enhancing the selection process by considering their interactions. The transferability of SHED-refined datasets is the key advantage over other methods. Additionally, While classic Shapley value calculation is impractical, SHED significantly reduces computational costs by utilizing an approximation method and only calculating Shapley values for cluster proxies rather than individual data points, making it both efficient and scalable. Flexible framework design gives SHED the potential to be applied to various tasks. **We believe that utilizing well-established methods as a part of SHED should not be grounds for rejection.**
> > >
> > > 3. Importance of Instruction Selection
> > > - **The need for instruction-tuning data selection is particularly important as fine-tuning datasets grow increasingly large and complex.** For example, datasets like P3, which includes over 100 million examples, the Pile with 825 GB of text, and LAION-400M with 400 million image-text pairs, illustrate the scale at which fine-tuning can occur. Efficient, scalable methods like SHED are essential for managing such large datasets. **Besides, as the theoretical and experimental analysis shows, our approach is designed to scale to much larger datasets, proving its value in broader applications.**
> > > - **Furthermore, most researchers lack the resources to support pre-training from scratch, making fine-tuning an essential strategy.** SHED-selected datasets have demonstrated strong performance across multiple models, and releasing these curated datasets to the community could provide a significant contribution, helping researchers save substantial resources.
> > > - **The field of instruction-tuning data selection may indeed be crowded, but the crowdedness does not diminish the value of contributions that address real and pressing challenges. The computational advantages, transferability, and flexible design of SHED make it a valuable tool in this area.** Moreover, as instruction-tuning datasets continue to grow in size and complexity, the need for efficient, scalable data selection methods like SHED will only increase. By providing a framework that is both efficient and adaptable, we believe SHED makes a meaningful contribution to the field, with potential applications extending well beyond the current scope.

---

> > > > ### Comment · Area_Chair_boSB · 2024-08-13
> > > >
> > > > Reviewer tj1t, do you have any additional questions or feedback?

---

> > > > > ### Comment · Reviewer_tj1t · 2024-08-13
> > > > > **Thanks for the prompt response**
> > > > >
> > > > > Thanks to the authors for the prompt response. My review remains unchanged.

---

### Official Review · Reviewer_Df9d · 2024-07-16

**Soundness:** 3
**Presentation:** 2
**Contribution:** 2
**Rating:** 6
**Confidence:** 2

**Summary:**

The paper proposed a data refinement framework that refines datasets for fine-tuning LLMs by using the Shapley value. Based on the description, SHED is able to create smaller, high-quality datasets from large, extensive datasets without human intervention or commercial LLMs. This process involves three key components: model-agnostic clustering, proxy-based Shapley calculator, and optimization-aware sampling. Extensive experiments show that datasets curated by SHED achieve comparable or superior performance to full datasets while significantly reducing the data size and computational costs. Moreover, SHED-curated datasets exhibit strong transferability across various LLMs.

**Strengths:**

1. The studied problem is critical as the proposed method is general to creat small yet high-quality datasets that benefit LLM finetuning in terms of performance improvement and computational cost reduction.
2. The experiments are extensive by covering multiple datasets and fine-tuning tasks.

**Weaknesses:**

1. While SHED reduces the computational complexity of calculating Shapley values based on clustering and proxy-based calculations, it is unclear how effecient (e.g., in terms of time) it is compared to the classical calculation method.
2. As authods illustrated, the reliance on clustering may inadvertently reduce data diversity, which results in the overlook of rare but important samples.

**Questions:**

1. Is it possible to evaluate SHED on a larger dataset in the industry level considering that the used datasets in the paper are still relatively small?
2. How will the selection of initial clustering groups affect the final model performance?

**Limitations:**

No potential negative societal impact is observed.

---

> ### Author Rebuttal · Authors · 2024-08-06
>
> We’d like to thank the reviewer for the insightful and positive feedback. We are encouraged that the reviewer found our work meaningful, novel, and effective. For the thoughtful questions and constructive suggestions. We'd like to share our responses below.
>
> **W.1:**
>
> We appreciate the reviewer's observation regarding the efficiency of SHED compared to classical Shapley value calculation methods. In our paper, **we provide a detailed analysis of SHED's computational complexity and actual runtime. As shown in Figure 3 (c), we present the computational time for Shapley value calculations across different cluster numbers. This graph clearly illustrates how SHED's runtime scales with the number of clusters, providing concrete evidence of its computational efficiency.**
>
> In contrast, for the classical Shapley value calculation method, the time complexity is exponential in the size of the dataset. Given our dataset size is about 100,000 instances, this results in a computational cost that is prohibitively large. As a result, it is not feasible to provide actual runtime measurements for the classical method on our dataset. However, we can analyze the efficiency of SHED compared with the classic Shapley value method under the experimental settings of this paper.
>
> **For the classical Shapley value calculation** method with a dataset of |D| = 100,000 instances: We need to consider all possible subsets (2^|D|). For each subset S, we fine-tune the model (taking time |S|t, where |S| is the size of the subset) and evaluate it on the test set (taking time T_m). Therefore, **the time complexity would be:
> O(2^|D| * (|D|t/2 + T_m)) = O(2^100,000 * (50,000t + T_m))
> This is computationally infeasible.**
>
> In contrast, **SHED's complexity, as detailed in our paper’s discussion section, is:
> O((Ck/n)[(C+n)t/2 + T_m]) = O(500 * [1530t + T_m]).
> This demonstrates a significant reduction in computational complexity compared to the classical method.**
>
> **W.2:**
>
> We appreciate the reviewer's insightful observation regarding the potential limitation of our clustering approach in SHED.
>
> **Our use of clustering and proxy data points is a trade-off we make to significantly reduce computational complexity while still capturing the overall data distribution effectively.**
>
> However, **we have taken steps to mitigate the overlooking issue.** The Quality-Weighted Cluster Sampling (QWCS) variant of our method allows for more diverse sampling. As shown in our experiments, despite this potential limitation, SHED still outperforms other methods across various tasks, suggesting that **the benefits of our approach outweigh this drawback in practice.**
>
> Nevertheless, we agree that this is an important area for improvement. In our future work, we plan to explore more sophisticated clustering algorithms and sampling strategies that could better preserve rare but important samples. We thank the reviewer for highlighting this important aspect, which will help us improve our method.
>
> **Q.1:**
>
> We appreciate the reviewer's question about evaluating SHED on industry-level datasets. We'd like to address this point:
>
> - **Scalability:** SHED's design, particularly its use of clustering and proxy data points, makes it **inherently scalable to larger datasets. By setting the ratio C/n to a fixed value, as done in the experimental setup, the computational complexity of SHED grows linearly with the number of clusters**, not exponentially with the dataset size like traditional Shapley value calculations.
>
> - **Current Limitations:** Our current experiments were constrained by the computational resources available to us in an academic setting. Our current computational resources are limited and insufficient to conduct experiments on industry-scale datasets within a short timeframe.
>
> In the future, **we plan to rent additional server resources to apply SHED on much larger datasets. We are committed to open-sourcing the curated datasets from these experiments, contributing valuable resources to the research community. These efforts will further validate SHED's effectiveness at industrial scales and support broader research.**
>
> We thank the reviewer for this suggestion, allowing us to emphasize our commitment to contributing to the research community despite current resource constraints.
>
> **Q.2:**
>
> We appreciate the reviewer's insightful question about the impact of initial clustering on SHED's performance.
> The initial clustering step in SHED plays a significant role as it determines the proxy data points used for Shapley value calculation. Our method employs several strategies to ensure robustness to variations in initial clustering:
>
> - **Semantic-preserving embeddings:** We use Sentence Transformers to generate embeddings, which effectively capture semantic similarities across various domains. This helps ensure that semantically similar data points are likely to be clustered together, even if the exact cluster boundaries may vary.
>
> - **Multiple initializations:** In fact, we conducted multiple experiments with different random initializations to verify the robustness. **We found that the number of clusters is a factor that affects the performance.** Therefore, we conducted a sensitive analysis of the number of clusters. Our sensitivity analysis (Figure 3) shows that performance improvements plateau when the number of clusters exceeds 3√|D|. This suggests that **as long as we choose a sufficiently large number of clusters, the exact choice is less critical.**
>
> - **Shapley value calculation:** This step considers the contribution of each proxy data point across multiple subsets, which helps to average out some of the variability introduced by clustering.
>
> - **Optimization-aware sampling:** The final selection step considers the Shapley values of all clusters, providing another layer of robustness against individual cluster variations.
>
> We thank the reviewer for this valuable question, which helps us clarify the robustness of SHED.

---

> > ### Comment · Reviewer_Df9d · 2024-08-12
> >
> > Thanks for the authors' response. They partially address my question related to the computation time of SHED. I have updated my scores.

---

### Official Review · Reviewer_kWYC · 2024-07-28

**Soundness:** 3
**Presentation:** 4
**Contribution:** 3
**Rating:** 6
**Confidence:** 3

**Summary:**

Tuning large language models to domain tasks is a difficult challenge requiring a dataset of high quality examples. Since noisy examples can significantly degrade performance, it is important to be able to curate these small, high-quality fine-tuning datasets. The authors propose filtering using each data point’s estimated contribution to the model performance using Shapley values. The proposed SHED method clusters the data, then iteratively deletes clusters based on the Shapley values of representative samples until a small dataset remains.

**Strengths:**

- The topic of the paper is well-motivated. It is known that large models need high quality data during the fine-tuning process, and are sensitive to noisy data in this stage. High quality data can be prohibitively expensive to collect
- Shapley values are a known and well-studied method to asses the importance of training points to a model’s predictions
- Unlike previous data selection methods employing Shapley values (TS-DSHAPLEY), shed uses clustering and representative sample selection to significantly reduce the computational cost of data subsection. SHED also considers different target objectives (e.g. fairness), although it is not clear how easy it would be to adapt other methods to this objective. This is an interesting novelty that can help make shapley values more practically useful in this data selection setting.
- Experiments are conducted on good language model eval datasets (e.g MMLU)
- Results show that datasets filtered via SHED achieve improved performance compared to other sampling methods, and also compared to full dataset fine-tuning
- Results are shown across a variety of evaluation datasets.

**Weaknesses:**

- The related work leave room for improvement. In particular, the primary focus of the paper is a method that uses the estimated effect of each data point on accuracy to filter data points. Yet, the discussion of such related methods is lacking. Some recent works (e.g. DsDm https://arxiv.org/abs/2401.12926) also use the impact of individual data points to subselect datasets.
- SHED uses a proxy data point (closest to the centroid) to estimate the predictive influence of the entire cluster. While this save on significant computational cost, the effectiveness of this stage depends significantly on the quality of the text embeddings used to compute similar, as well as the size and number of clusters
- The experimental method training leaves room for improvement. All methods are stated to train for the same number of epochs and same other hyperparameters. However, as training can be sensitive to hyperparameters, it could make sense to conduct a sweep based on some val set for each method, and present results for the best hyperparameters on the test set.
- It is not clear how the filtered data size was chosen. A variety of tables (e.g .Table 6 and 7) use different counts of QOCS and QWCS of 10k and 13k. Why not use the same ksubselected data size for all method for a clearer comparison?

**Questions:**

Why are the different data selection methods compared with different final selected dataset sizes?
How were the training hyperparameters chosen?

**Limitations:**

The authors have sufficiently discussed the limitations of their work.

---

> ### Author Rebuttal · Authors · 2024-08-06
>
> We thank the reviewer for the valuable feedback. We are encouraged by the positive comments on our well-motivated, clearly presented, and highly effective work. Below are our responses to the thoughtful questions and constructive suggestions:
>
> **W.1:**
>
> We sincerely thank the reviewer for highlighting this issue and the reference to DsDm. We acknowledge that our original manuscript overlooked DsDm, which shares similarities with our approach in using the impact of data points for selection. However, like other works, DsDm does not consider the combinatorial effects of data points in the same way that SHED does. The key differences are:
>
> - Individual Impact: DsDm primarily focuses on estimating the impact of individual data points on model performance. It uses data models to approximate how each training example affects the model's predictions on target tasks.
>
> - Linear Approximation: DsDm's data models are typically implemented as linear functions, which implicitly assume that each data point's contribution is independent of other points in the dataset.
>
> In contrast, SHED's use of Shapley values allows it to consider the marginal contribution of data points, **thereby capturing potential interaction effects between data points**.
>
> We will revise our manuscript to include a detailed discussion of DsDm and clearly articulate SHED's unique contributions.
> We appreciate this feedback, which will significantly improve our paper's comprehensiveness and scholarly value.
>
> **W.2:**
>
> We appreciate the reviewer's observation regarding SHED's use of proxy data points. Indeed, the effectiveness of this approach is influenced by the quality of embeddings and clustering parameters. We'd like to address these points:
>
> - **Sensitivity Analysis:** As detailed in our paper, we have already conducted a sensitivity analysis on the number of clusters. Figure 3 demonstrates how performance and computational time vary with different cluster numbers. This analysis shows that performance improvements plateau when the number of clusters exceeds 3√|D|, providing empirical guidance for choosing this parameter.
>
> - **Embedding Quality:** We use Sentence Transformers for generating embeddings, which are known for their ability to capture semantic similarities between sentences. By clustering on these embeddings, we group semantically similar text, which helps in discovering and organizing potential patterns and relationships in the text data, enhancing the representativeness of our proxy points.
>
> While our current approach has shown robust performance, we acknowledge the importance of this aspect. In our future work, we plan to further investigate the impact of different embedding techniques and clustering methods on SHED's performance. We thank the reviewer for this valuable feedback, which will help us improve our work.
>
> **W.3:**
>
> We appreciate the reviewer's insightful comment on our experimental methodology. We'd like to clarify our approach and reasoning:
>
> **Fairness in Comparison and Focus on Data Impact:** Our decision to use the same hyperparameters across all methods was deliberate, aiming to ensure a fair comparison that isolates the impact of data selection on model performance. This approach allows us to directly observe how different data selection methods affect performance without the confounding influence of varying hyperparameters.
>
> We acknowledge that our current approach of using fixed hyperparameters across all methods may not fully capture the optimal performance of each method.
>
> To address this concern, we propose the following improvements for our revised manuscript:
>
> - Hyperparameter Sweep: We will conduct a hyperparameter sweep for each method, including learning rate, batch size, and number of epochs.
>
> - Best Configuration Reporting: We will report results on the test set using the best hyperparameters found for each method during the sweep.
>
> We believe these additions will strengthen our experimental methodology. We thank the reviewer for this suggestion, which will enhance the reliability of our results.
>
> **W.4:**
>
> We appreciate the reviewer's observation regarding the varying data sizes used in our experiments. We acknowledge that this may have caused some confusion in interpreting our results. To address this concern:
>
> - **Comprehensive Comparison:** To ensure a fair comparison, **we have included results for a fixed data size of 10k samples across all methods in Table 2**. Additionally, as detailed **in Appendix B of our paper, we have conducted experiments comparing different methods across various sample sizes**. These provide a comparison of method performance when using identical amounts of data.
>
> - **Method-Specific Optimization:** In Appendix B, we provide comprehensive comparisons of different methods across various sample sizes. Our analysis reveals that QOCS often performs better with smaller subsets. This is likely because it prioritizes the highest quality, most informative data, making it particularly effective when working with limited data. On the other hand, QWCS often benefits from slightly larger subsets. This method balances quality with diversity, allowing for the inclusion of a broader range of data patterns. As a result, QWCS can capture more comprehensive representations of the dataset. These characteristics explain the different optimal subset sizes for each method. By presenting the best-performing subsets in the main text, we aim to demonstrate the full potential of each approach.
>
> We will revise our manuscript to more clearly explain our rationale for presenting the best-case scenarios and to direct readers to the appendix for these fixed-size comparisons.
>
> We thank the reviewer for this feedback, which will help us improve the clarity of our results presentation.
>
> **Q1:**
>
> Please see the responses R.3 and R.4. We have also presented all the training hyperparameters in Appendix C.

---

> > ### Author Response · Authors · 2024-08-13
> > **Looking forward to your response**
> >
> > Dear Reviewer,
> >
> > As we are approaching the end of the rebuttal period, we would like to cordially inquire about the extent to which we have successfully addressed the concerns outlined in your review.
> >
> > Should any lingering points require further attention, please rest assured that we are enthusiastic about the opportunity to provide comprehensive responses to any subsequent queries or comments you may have.
> >
> > Your constructive input remains invaluable to us, and we appreciate your dedication to enhancing the quality of our manuscript. Thank you for your time and consideration.
> >
> > Best, Authors

---

> > > ### Comment · Area_Chair_boSB · 2024-08-13
> > >
> > > Reviewer kWYC, do you have any additional questions or feedback?

---

> > ### Comment · Reviewer_kWYC · 2024-08-14
> > **Thank you for your response.**
> >
> > Thank you for your detailed response to my feedback. Please find an additional note below.
> >
> > W.1: Although Shapley does help consider interactions between data points, since you're re-computing the values after every cluster removal I don't think it actually matters here, i.e. compared to models that consider the contribution of each data point independently. Further, the claim that you are "capturing potential interaction effects between data points" is also not quite accurate, as the data points are grouped into larger clusters over which the Shapley values are computed.

---

> > > ### Author Response · Authors · 2024-08-14
> > > **Thank you for the feedback**
> > >
> > > Thank you for your insightful comment. We would like to address your concerns regarding the interaction effects between data points.
> > >
> > > The primary objective of removing clusters in our method is to calculate the marginal contribution, or boundary effect, of each cluster relative to the remaining clusters. By sequentially removing clusters and observing their impact on model performance, we can get how each cluster contributes to the overall model accuracy. After calculating the boundary contribution of a cluster in one iteration, we restart the process with a new randomized removal order. This repetition is crucial because it allows us to capture the variation in each cluster (or the data points within them)'s marginal contribution across different combinations of clusters. **By performing this process multiple times, with different sequences of cluster removal, we can average the different marginal contributions to accurately estimate the true Shapley value for each cluster. This allows us to observe how the presence or absence of a cluster in different cluster combinations influences the overall model performance, thereby capturing the interaction effects between the data points in different clusters.**
> > >
> > > **We agree with that the claim about capturing "interaction effects between data points" might not have been entirely accurate. It is more precise to state that our method considers the interaction effects between data points in different clusters. The clustering process itself is designed to group data points that are likely to interact in similar ways with the model and data points in other clusters.** By using Shapley values on these representative clusters, we capture a higher-level interaction effect, which is crucial for scaling the computation to larger datasets. While it’s not the same as evaluating every individual interaction, **this method strikes a balance between computational feasibility and the need to preserve interaction effects,** which we believe adds value to the data selection process.
> > >
> > > We sincerely appreciate your insightful feedback and will revise our manuscript to accurately reflect this aspect of our method. Your feedback is incredibly helpful in improving our paper.

---

### Public Comment · ~Brando_Miranda1 · 2025-03-31
**Friendly Request for Reproducible Code**

Hi Authors,

I congratulate you on such interesting work! I really appreciated the idea of considering how examples interact (the main idea of your paper!) when selecting data.

I was wondering, may I request a reproducible version of your code that is easy to run please?

Thank you for considering my request.

Sincerely, Brando Miranda

PS: a quick Google search with "SHED: Shapley-Based Automated Dataset Refinement for Instruction Fine-Tuning Code" and related key words / phrases does not help me find a previous version of the code I once used I think. May I request the title of the repo be change to something more easily searchable please?

--- Edit

PS2: I found a link to the repo! https://github.com/Lucidreamer9/SHED-Shapley-Based-Automated-Dataset-Refinement/ Not sure why it was so hard to find.

---

> ### Public Comment · ~Yexiao_He1 · 2025-03-31
> **The code of SHED**
>
> You can find the code and dataset here:
> https://github.com/Lucidreamer9/SHED-Shapley-Based-Automated-Dataset-Refinement

---

### Decision · Program_Chairs · 2024-09-25

**Decision:**

Accept (poster)

**Comment:**

The reviews for this paper are mixed, with 2 weak accept ratings and one reject. There was active discussion between the reviewer recommending rejection and the authors, but the reviewers concerns were not sufficiently addressed to lead to an improved score. The main concerns raised were the significant reliance of the method on the existence of a sufficiently representative embedding of the data, and the similarity of the method to previous ones such as TS-DSHAPLEY. The authors emphasize that their method outperforms the previous, similar work. After reading the paper, the reviews, and the author's responses, and particularly the reviewer leaning for rejection's own admission that they could not find "a particular, critical flaw in the paper" the ACs agree that the paper has made enough of a contribution to recommend acceptance. However, the heavy reliance on the existence of a sufficiently representative embedding is a significant limitation to the value of the contribution to many real-world challenges where such embeddings do not exist, and this should be captured in more detail as a limitation of the work in the paper.